# Diagnostic value of urinary and serum IgG antibodies in evaluating drug treatment response in strongyloidiasis assessed by fecal examination and digital droplet PCR

Phattharaphon Wongphutorn[1,2], Kulthida Y. Kopolrat[3], Chanika Worasith[4], Chatanun Eamudomkarn[5]*, Nuttanan Hongsrichan[5], Opal Pitaksakulrat[5], Jiraporn Sithithaworn[6], Patcharaporn Tippayawat[6], Anchalee Techasen[6], Rahmah Noordin[7], Thomas Crellen[8], Paiboon Sithithaworn[2,5]*

1 Biomedical Science Program, Graduate School, Khon Kaen University, Khon Kaen, Thailand, 2 Cholangiocarcinoma Research Institute, Khon Kaen University, Khon Kaen, Thailand, 3 Faculty of Public Health, Kasetsart University Chalermphrakiat Sakon Nakhon Province Campus, Sakon Nakhon, Thailand, 4 Department of Adult Nursing, Faculty of Nursing, Khon Kaen University, Khon Kaen, Thailand, 5 Department of Parasitology, Faculty of Medicine, Khon Kaen University, Khon Kaen, Thailand, 6 Department of Medical Technology, Faculty of Associated Medical Sciences, Khon Kaen University, Khon Kaen, Thailand, 7 Department of Parasitology and Medical Entomology, Faculty of Medicine, Universiti Kebangsaan, Kuala Lumpur, Malaysia, 8 School of Biodiversity One Health and Veterinary Medicine, Graham Kerr Building, University of Glasgow, Glasgow, United Kingdom

* paib_sit@kku.ac.th (PS); chatea@kku.ac.th (CE)

## Abstract

Detection of *Strogyloides*-specific IgG antibodies in urine and serum has been used in diagnostic and epidemiological studies on strongyloidiasis. However, the usefulness of these assays in assessing responses to anthelmintic treatment is unclear. Thus, we evaluated the diagnostic performance and temporal profiles of *Strongyloides*-specific IgG antibodies in a cohort of participants at baseline and post-treatment. The participants were prospectively screened for baseline parasitic infections by fecal examination [agar plate culture technique (APCT) and formalin-ethyl acetate concentration technique (FECT)] and digital droplet polymerase reaction (ddPCR) for *Strongyloides stercoralis*. At each sampling point, *Strongyloides*-specific IgG in urine and serum were measured by an in-house *S. ratti*-based enzyme-linked immunosorbent assay (ELISA). At baseline, 169 of 351 participants (48.1%) had *S. stercoralis* infection by the combined fecal examination and ddPCR. The diagnostic sensitivities of IgG in urine and serum were 91.1% and 88.2%, respectively. The participants were given treatment with a single oral dose of ivermectin (IVM, 200 µg/kg) and were followed up by fecal and immunological diagnosis at 3 to 18 months post-treatment. The cure rate of IVM treatment evaluated by APCT and ddPCR was 88.3% at three months post-treatment. The profiles of IgG in urine in the curative treatment group showed a significant trend of decline with time post-treatment (Kruskal-Wallis test = 113.4–212.6, *p* value < 0.0001) and the lowest levels were seen 12 months post-treatment. The treatment response (> 50% reduction in urinary IgG antibody units) was 100%, and conversion from positive to negative results was 65.4%. The treatment response and conversion to negative assessed

**Data Availability Statement:** All relevant data are within the manuscript and its Supporting information files.

**Funding:** 1.National Research Council of Thailand (NRCT) 2.Basic Research Fund of Khon Kaen University through Cholangiocarcinoma Research Institute (CARIBRF64-50). The funders had no role in study design, data collection and analysis, decision to publish, or preparation of the manuscript.

**Competing interests:** The authors have declared that no competing interests exist.

by serum IgG-ELISA were similar to those by urine IgG-ELISA. The results from this long-term diagnostic study highlight the utility of urinary IgG and serum IgG for screening and monitoring treatment outcomes in strongyloidiasis.

## Introduction

Strongyloidiasis, a Neglected Tropical Disease caused by *Strongyloides stercoralis*, is a global health concern with a worldwide distribution, particularly in Southeast Asia, including Thailand, Lao PDR, and Cambodia [1–5]. This condition, often asymptomatic, can lead to life-threatening complications if corticosteroid treatment is involved or the patient is immunosuppressed, leading to hyperinfection and dissemination. Hence, the importance of accurate diagnosis and treatment of *S. stercoralis* infection cannot be overstated in patient management.

Fecal examination is the standard approach for diagnosing *S. stercoralis* infection, such as the agar plate culture technique (APCT). However, it has low sensitivity (23.0–30.8% when measured against a composite standard) and high daily variability [6]. Multiple fecal examinations over three consecutive days are recommended to increase the sensitivity, but this is inconvenient for patients, laboratory staff, and healthcare workers [6, 7]. Alternatively, studies using copro-DNA detection by polymerase reaction (PCR) or Droplet Digital™ PCR (ddPCR) have reported a higher sensitivity and specificity [8–11].

With regard to immunological methods, the detection of specific IgG antibodies in serum, saliva, and urine have been reported with varying sensitivities [8, 12–15]. Also, serum-specific IgG has been used to evaluate outcomes of ivermectin (IVM) administration in terms of treatment responses (antibody reduction $\geq$ 50%) and negative seroconversion (negativization) [16].

Unlike blood collection, urine collection is a less invasive and widely accepted sampling method, which could potentially enhance accessibility to diagnostic testing and improve screening efforts for helminths which have previously relied on fecal examination [14, 17]. The detections of specific IgG in urine have shown promising results, with a comparable sensitivity of 81.1–88.4% by either *S. ratti*, *S. stercoralis* or NIE-ELISA [13]. We propose that urinary IgG could be as effective as serum IgG in monitoring IVM treatment responses and negative seroconversion in strongyloidiasis.

In this study, we evaluated the usefulness of urinary IgG in comparison with serum IgG as an infection immunological marker and the response to curative treatment by IVM. The outcomes of anthelmintic treatment were analyzed using a comprehensive diagnostic approach, combining parasitological and molecular methods with serological analysis of urine and serum.

## Materials and methods

### Ethics statement

The human subject protocol used in this study was approved by the Human Ethics Committee of Khon Kaen University, Thailand (reference number: HE654013) according to the principles expressed in the Declaration of Helsinki. Written informed consent was obtained from individual participants. The project was from 20 June 2022 to 19 May 2023 and from 22 July 2023 to 29 January 2024. At the end of the study, all parasite-infected participants detected during the project operation were given specific anthelmintic treatments. All samples were anonymized, and blinding was applied during the analysis and result interpretation.

The laboratory animal protocol for maintaining *S. ratti* in Wistar rats for antigen production was approved by the Institutional Animal Ethical Committee, Khon Kaen University (IACUC-KKU-99/62). Animal handling and husbandry were performed strictly per guidelines for the Care and Use of Laboratory Animals of the National Research Council of Thailand.

## Study area, project participants and clinical samples

This prospective cohort study was conducted in Phon Ngam subdistrict, Kamalasai district, Kalasin province in northeast Thailand. The project involved screening for *S. stercoralis* infection and following up after IVM treatment using parasitological, molecular, and immunological methods. The participants' eligibility criteria for recruitment and study inclusion were: (i) native residents of the area; (ii) all genders aged > 15 years; (iii) consent to provide clinical samples (feces, blood, and urine); (iv) in good general health and with no clinical signs or symptoms of liver or kidney disease. The sample size of this study was calculated based on the proportion of the population expected to be positive for *S. stercoralis*, i.e., 28% [18] with the standard score (Z-score) set at 1.96, corresponding to a 95% confidence level ± 5%. The calculated sample size was 310 participants. We expected up to 30% loss to follow-up and incomplete donation of clinical specimens; therefore, the final estimated sample size was 400 participants.

Of the 437 participants initially recruited, 86 were excluded due to failure to submit complete clinical specimens. Three hundred fifty-one participants provided written informed consent and completed clinical specimens (feces, blood, and urine) (Fig 1). The participants were separated into three groups: Group 1 individuals were *S. stercoralis*-positive by fecal examination and/or ddPCR or *Strongyloides* positive by reference test (n = 169). Group 2 individuals were infected with other parasite infections, not *S. stercoralis* (n = 70). Group 3 consisted of parasite-negative individuals (n = 44).

Recruited participants were requested to provide clinical samples at the baseline screening and follow-ups at five-time points, i.e., 3, 6, 9, 12 and 18 months post-treatment. Ten grams of fecal samples from each participant were collected in a clean, wide-mouth container at each sampling point. Each fecal sample was separated into aliquots for analysis by APCT, FECT, and ddPCR. However, the ddPCR was performed at three of the five follow-up time points, i.e., 3, 12, and 18 months post-treatment. For the urine sample, 10 milliliters of first-morning urine were obtained from each participant and were preserved in a tube containing 0.1% $NaN_3$ as preservative. Each urine sample was transferred into a microwell plate and kept at 4°C until required. Five milliliters of blood from each participant were collected in 10 ml serum separating tubes (Clot activator tube, BD, United Kingdom). Aliquots of 500 μl were transferred to 1.5 ml microcentrifuge tubes. The levels of IgG in the serum and urine samples were measured by *S. ratti*-based ELISA. All sample analyses were anonymized, and blinding was applied during analysis and result interpretation.

## Fecal examination methods

The fecal examination methods used to assess parasitic infections were FECT and APCT. In the former, two grams of formalin-fixed samples were filtered through gauze, and ethyl acetate was used to extract fat from feces. After centrifugation at 1,455 x *g* for 5 mins, the supernatant was removed, and the sediment suspended in 1 mL of 10% formalin. Three drops of the final fecal suspension were sampled and examined by microscopy. The number of larvae per gram of feces was calculated [19].

The APCT as a reference standard for detecting *S. stercoralis* was performed according to Koga et al. [20]. Approximately 4 grams of feces were smeared on a 1.5% nutrient agar plate in

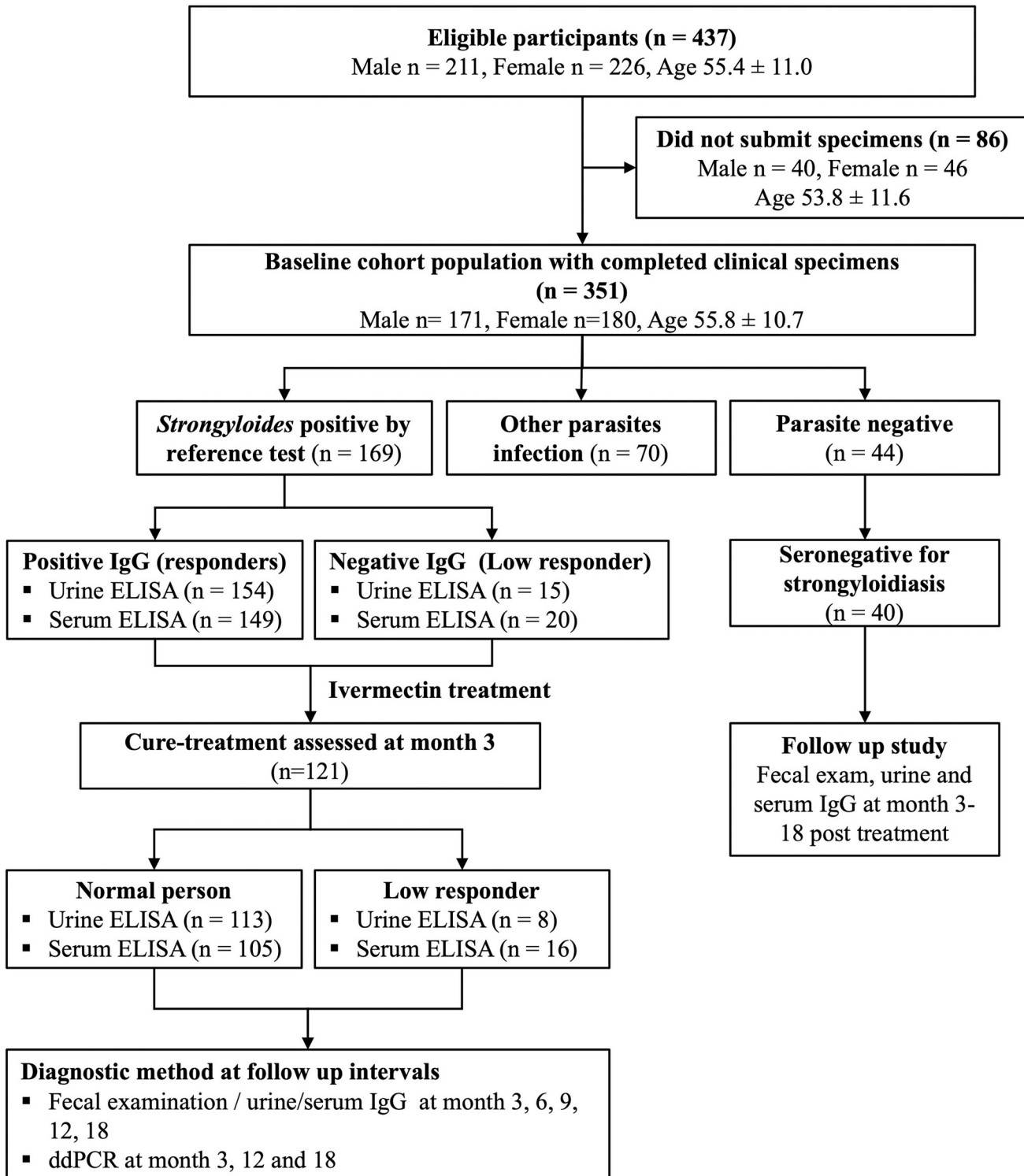

**Fig 1. The flow of the study and number of participants at each stage, from baseline screening to post-treatment follow-up.**

a plastic petri dish and incubated at 25˚C for seven days, and the presence of worms was assessed every day from day 2 to 7. The presence of larvae was examined under a stereo microscope before washing the surface of the lid and plate with 10% formalin and transferring it into a centrifuge tube. After centrifugation, wet sediment preparation was examined by a light microscope to differentiate between *S. stercoralis* and hookworm.

## Molecular diagnosis by droplet digital PCR

Fecal DNA extraction was performed using the QIAmp PowerFecal® Pro DNA kit (QIAGEN, USA) according to the manufacturer's instructions. First, 250 mg of fecal sample and 800 µL of solution CD1 were added to a PowerBead Pro tube and vortexed for 10 minutes to lyse all cells. DNA was eluted using 50–100 µL of solution C6, centrifuged at 15,000 x *g* for 1 min, then stored at -20˚C.

The ddPCR (Bio-Rad Laboratories, Hercules, CA) was used to detect *S. stercoralis* DNA in fecal samples using the published protocol [11]. The PCR reaction comprised 20 µl of 2X ddPCR Supermix for probes without dUTP (Bio-Rad), 250 nM specific primer (forward 5′– CCGGACACTATAAGGATTGA–3′, reverse 5′ ACAGACCTGTTATCGCTCTC–3′) labelled with FAM probe (5′–TCCGATAACGAGCGAGACTT–3′) which is specific for 18s rRNA gene of *S. stercoralis* (Bio-Rad), RNase/DNase free water, and DNA template. For droplet generation, the reactions of the different samples were added to separate wells of DG8 cartridge (Bio-Rad), and 70 µL of droplet generation oil was loaded into the bottom wells. A single droplet was generated using a QX200 droplet generator (Bio-Rad). The thermal cycling protocol was set up as described [11]. The PCR product was analyzed by QuantaSoft Software using QX200 droplet reader (Bio-Rad), and accurate data was obtained from 10,000 droplets. Finally, the concentration of target DNA and the number of total droplets were measured by automatic thresholding of ABS analysis. The cut-off of a negative and positive result was set up according to the threshold of positive *S. stercoralis* and non-DNA template control. It provided clear discrimination, with a manually adjustable threshold amplitude at 2,000 [11].

## Antigen preparation

The life cycle of *S. ratti* was maintained in Wistar rats [14, 17]. Infection of Wistar rats was performed by subcutaneous inoculation of third-stage filariform larvae (5,000 L3/rat). The larvae were harvested from the culture of the rats' fecal samples by filter paper technique, and the antigen was prepared by sonicating washed larvae [21]. The Bradford protein assay (Pierce™ BCA Protein Assay Kit, Thermo Scientific, USA) was used to determine the antigen's total protein concentration.

## Measurement of specific IgG by ELISA

The unit-based ELISA protocol described previously was used with slight modifications [14, 17]. The 96-well plates were coated with 2.5 µg/mL of crude *S. ratti* antigen and kept at 4˚C overnight. The plates were washed twice with phosphate saline buffer (PBS, pH 7.2) containing 0.05% Tween-20 before blocking with 3% skim milk in PBS containing 0.5% Tween-20 at ambient temperature for 2 hrs. One hundred microliters of undiluted urine or diluted serum samples (1:8000) were added and incubated at 37˚C for 1 hour. After the plates were washed three times, 100 µL of horseradish peroxidase conjugate goat antihuman IgG (dilution at 1:4000) (Abcam Inc., UK) was added and incubated at 37˚C for 1 hr. Then, the plates were washed three times before adding o-phenylenediamine (OPD) substrate (Sigma, St. Louis, MO, USA) in citrate phosphate buffer (pH 5.0) at 100 µL/well and incubated at room temperature for 1 hour. Finally, the reaction was stopped by adding 4 N sulfuric acid

50 μL/well, and the optical density (OD) was measured at 492 nm by an ELISA reader (TECAN Sunrise, Austria). The OD was transformed into an antibody unit by comparing it to a standard curve obtained from three-fold serial dilutions of pooled positive serum samples [14].

## Anthelminthic treatment and follow-up

The participants diagnosed with *S. stercoralis* infection by fecal examination or ddPCR were treated with 200 μg/kg IVM by a single oral dose administration [22]. The subjects infected with other trematode and cestode parasites were treated with a single oral dose of praziquantel (40 mg/kg) at the end of the study [23–25]. At baseline and follow-up studies, the participants' urine and serum samples were collected for IgG measurements using *S. ratti*-based ELISAs. The cure rate of strongyloidiasis was evaluated using fecal examination and ddPCR. The levels of IgG at the baseline and at 3–18 months post-treatment were compared to assess the treatment response and the conversion to negative rates. The criterion of treatment response was based on the antibody reduction of at least 50% between baseline and post-treatment. The conversion to negative (negativization) is a transition of positive to negative results by any diagnostic test, including seroconversion of the urine or serum immunoassays [16].

## Statistical analysis

The statistical analyses of data obtained in this study were performed using SPSS v.26.0. The prevalence profiles of *S. stercoralis* classified according to age and gender were analyzed by the Chi-square test. The serum and urinary IgG antibody unit data was normalized using log-transformed data and log (1+ unit) before performing statistical tests. The intensity of *S. stercoralis* infection was measured using larval count, and the IgG antibody unit was analyzed by non-parametric tests (Man-Whitney U and Kruskal-Wallis test).

The cutoff values for IgG in urine and serum were determined by receiver operating characteristic (ROC) analysis using Medcalc version 11.6.1.0 based on analysis of 40 proven-positive and 40 proven-negative sera samples for *S. stercoralis*. The area under the curve (AUC) indicates how well a parameter can be differentiated between two diagnostic groups and the cutoff was calculated for the highest sensitivity and specificity [17, 26, 27]. The IgG cutoff values (antibody units/mL) were 140.7 [log (antibody unit +1 = 2.15)] in urine and 174.4 [log (antibody unit +1 = 2.24)] in serum.

Detection rates or diagnostic sensitivity of IgG in urine and serum were calculated using the combined results of APCT, FECT, and ddPCR as a composite reference standard. The sensitivity was calculated using Medcalc version 11.6.1.0. Pairwise comparisons of AUC values were evaluated by the DeLong's test [28, 29]. Agreement of diagnostic methods was evaluated using Cohen's kappa coefficient (κ), for which a value of less than 0.2 is considered to indicate slight agreement, 0.21–0.40 is fair agreement, 0.41–0.60 is moderate agreement, 0.61–0.80 is substantial agreement and > 0.80 is perfect agreement [30, 31]. The correlation of *Strongyloides*-specific IgG in urine and serum was measured by the Spearman correlation test (r). The percent cure rate of strongyloidiasis was evaluated by comparing the results of fecal examination and ddPCR between baseline and 3 months post-treatment. The results of statistical tests were considered significant when the *p* value < 0.05. The time profiles of positive rates and levels of *Strongyloides*-specific IgG in urine and serum were validated by Chi square test for trend and Kruskal-Wallis test, respectively. The correction to the Type I error for multiple comparison was implemented to ensure the confidence in the reported results.

## Results

### Baseline study

**Characteristics of the study participants and baseline parasitic infections.** The number of participants who submitted complete clinical specimens for the baseline fecal examination was 351 (Table 1). The prevalence of *S. sterocalis* determined by APCT was 45.8% (124/271), 21.4% (75/351) by FECT and 45.3% (43/95) by ddPCR. The prevalence by composite reference tests (combined APCT, FECT and ddPCR) was 48.1% (169/351). The intensity of *S. stercoralis* infection estimated by FECT was 33.1 ± 7.9 larvae per gram feces and ddPCR was 74.7 ± 19.1 DNA copies/µL (geometric mean ± SE). Other parasitic infections were found in 108 of 351 individuals, and they included *Opisthorchis viverrini* (23.1%), *Taenia* sp. (4.8%), minute intestinal flukes (3.1%), hookworms (2.8%), *Echinostoma* spp. (0.9%), and *Trichuris trichiura* (0.9%).

**Prevalence and intensity of *S. stercoralis* infection by multiple diagnostic methods.** The prevalence of *S. stercoralis* by fecal examination (APCT or FECT) was 43.6% (153/351) and 48.1% by combined APCT, FECT and ddPCR. The prevalence of *S. stercoralis* based on urine IgG was 70.7%, and serum IgG was 68.4% (McNemar test = 84.3, $p$ value > 0.05). The prevalence of combined fecal examination and ddPCR (48.1%) was significantly lower than that of urine and serum IgG (McNemar test = 14.1–69.3, $p$ value < 0.001).

The prevalence and intensity of *S. stercoralis* infection in relation to age group are shown in Fig 2. The prevalence was significantly associated with age when determined by fecal examination (Chi-square test for trend = 9.8, $p$ value < 0.01), urine IgG (Chi-square test for trend = 27.7, $p$ value < 0.0001), and serum IgG (Chi-square test for trend = 13.1, $p$ value < 0.001). In both cases, the aggregated prevalence increases with age and peaks for participants > 70 years.

**Table 1. Characteristics of the study participants and parasitic infections based on fecal examinations (FECT and APCT), ddPCR and the composite reference standard test.**

| Variables | Values (%) |
|---|---|
| Age (Mean ± SD) | 55.8 ± 10.8 |
| Males | 171 (48.7%) |
| Females | 180 (51.2%) |
| Prevalence of parasites by FECT (n = 351) | No. positive (%) |
| • *S. stercoralis* | 75 (21.4) |
| • *O. viverrini* | 81 (23.1) |
| • *Taenia* sp. | 17 (4.8) |
| • Minute intestinal flukes | 11 (3.1) |
| • Hookworms | 10 (2.8) |
| • *Echinostoma* sp. | 3 (0.9) |
| • *Trichuris trichiura* | 3 (0.9) |
| Prevalence by FECT and APCT (n = 351) | |
| • *S. stercoralis* only | 116 (33.0) |
| • *S. stercoralis* and other parasites | 37 (10.5) |
| Prevalence of *S. stercoralis* by | |
| • APCT (n = 271) | 124 (45.8) |
| • APCT or FECT (n = 351) | 153 (43.6) |
| • ddPCR (n = 95) | 43 (45.3) |
| • Combined FECT, APCT and ddPCR (n = 351) | 169 (48.1) |

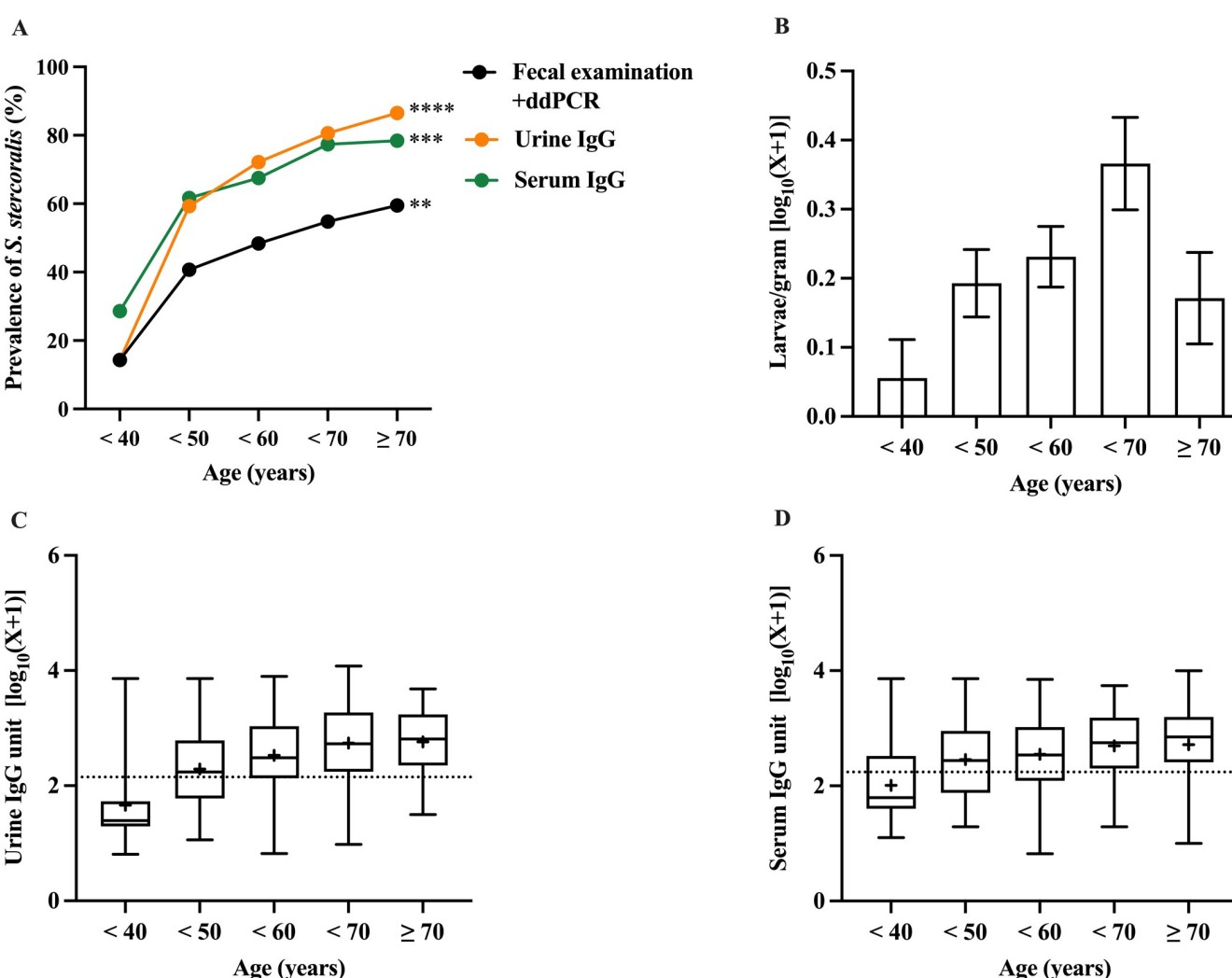

**Fig 2. Age-related prevalence and intensity of *S. stercoralis* infection by fecal examination and IgG antibody detection in urine and serum.** Prevalence by age (A), Intensity of infection by FECT [larvae per gram feces] (B), urine IgG antibody units (C), serum IgG antibody units (D). Dots show per cent positive rates, bar graphs with box whisker plots represent min-max with median and + symbol represented mean value.

The intensity of *S. stercoralis* infection measured by FECT showed an increasing trend up to 70 years of age, before declining in participants > 70 years. However, this trend with age was not found to be significant (Kruskal-Wallis test = 6.8, *p* value > 0.05) (Fig 2B). The concentration of specific-IgG antibody (unit/mL) in serum and urine were significantly elevated with age (Fig 2C and 2D, Kruskal-Wallis test = 18.9–37.6, *p* value < 0.0001, *p* value < 0.001, respectively).

The prevalence of *S. stercoralis* was significantly higher in males than females when determined by fecal examination and ddPCR (Chi-square test = 11.2, *p* value < 0.001), urine and serum IgGs (Chi-square test = 5.7–9.0, *p* value < 0.05 and *p* value < 0.01, respectively) (S1A Fig). A significant difference in intensity of *S. stercoralis* between genders was observed by FECT (Mann-Whitney test, *p* value < 0.0001) (S1B Fig) and IgG-ELISAs in urine and serum (Mann-Whitney test, *p* value <0.01 and *p* value < 0.05) (S1C and S1D Fig).

**Table 2. Positive detection rates of strongyloidiasis by urine and serum IgG in three groups of participants based on fecal examination and reference tests at baseline screening.**

| Population | n | Urine IgG (%) | Serum IgG (%) |
|---|---|---|---|
| 1. *Strongyloides* positive by reference test | 169 | 154 (91.1) | 149 (88.2) |
| 2. Other parasites infections | 70 | 27 (38.6)[a] | 43 (61.4) |
| 3. Endemic parasite-negative | 44 | 4 (9.1) | 4 (9.1) |
| **Total** | **283** | **185 (65.4)** | **196 (69.3)** |

[a] Significant difference between urine IgG- and serum IgG-ELISAs.

**Diagnostic performance and agreement.** The diagnostic performance of urine and serum IgG-ELISAs was evaluated using fecal examination and ddPCR as reference methods (n = 283). At baseline, the participants were divided into three groups (Table 2): Group 1 was *Strongyloides* positive by reference test who had *S. stercoralis* larvae or DNA in their feces (n = 169), Group 2 was individuals from *Strongyloides* endemic areas who had parasitic infections other than *S. stercoralis* (n = 70), and Group 3 was individuals from *Strongyloides* endemic areas who were negative for all parasites by fecal examination (n = 44). Diagnostic sensitivity of *S. ratti*-based IgG-ELISA in urine and serum were 91.1% and 88.2%, respectively (McNemar test = 12.5, *p* value = 0.405) (Table 3). The AUCs for urine IgG (AUC = 0.657, 95% CI = 0.589–0.724) and serum IgG (AUC = 0.727, 95% CI = 0.663–0.790) were significantly different (DeLong's test, *p* value < 0.01) (Table 3).

The diagnostic agreement between IgG-ELISAs (serum and urine) and fecal examination was assessed using Cohen's kappa coefficient (κ) (Table 4). It revealed a substantial agreement between fecal examination and urine IgG (κ = 0.645, 95% CI = 0.563–0743, *p* value < 0.001) and moderate agreement between fecal examination and serum IgG (κ = 0.488, 95% CI = 0.380–0.593, *p* value < 0.001). The agreement between urine and serum IgG-ELISAs was also moderate (κ = 0.575, 95% CI = 0.463–0.676, *p* value < 0.001).

**Table 3. Diagnostic sensitivity and accuracy of urine and serum IgG-ELISAs using combined fecal examination and ddPCR as a composite reference standard (n = 169).**

| Diagnostic methods | Diagnostic sensitivity[a] (%) (95% CI) | ROC analysis | |
|---|---|---|---|
| | | AUC (95% CI) | *p* value |
| Urine IgG-ELISA | 91.1 (85.8–95.0) | 0.657 (0.589–0.724)[b] | < 0.001 |
| Serum IgG-ELISA | 88.2 (82.3–92.6) | 0.727 (0.663–0.790) | < 0.001 |

[a] *Strongyloides* positive by reference test who had *S. stercoralis* larvae or DNA in feces (n = 169) as a composite reference standard

[b] Significant difference between urine IgG- and serum IgG-ELISAs.

**Table 4. Diagnostic agreements between diagnostic methods for strongyloidiasis by Cohen's Kappa analysis.**

| Paired diagnostic methods | n | κ value (95% CI) | *p* value | Agreement level |
|---|---|---|---|---|
| Fecal examination-Urine IgG | 283 | 0.654 (0.563–0743) | < 0.001 | Substantial |
| Fecal examination-Serum IgG | 283 | 0.488 (0.380–0.593) | < 0.001 | Moderate |
| Urine IgG-Serum IgG | 283 | 0.575 (0.463–0.676) | < 0.001 | Moderate |

**Correlation between specific IgG in urine and serum.** Among *Strongyloides* positive by reference test i.e. by fecal examination and ddPCR (n = 169), a statistically significant correlation was found between the level of IgG in urine and serum (r = 0.3348, *p* value < 0.0001) (S2 Fig).

## Follow-up study

**Cure rates of ivermectin (IVM) treatment.** Three months after IVM treatment of Group 1 participants (n = 169), the cure rate of *S. stercoralis* infection was evaluated based on the combined fecal examination and ddPCR. At 3 months post treatment, the results revealed positive tests for *S. stercoralis* infection in 16 of 137 treated individuals, thus the percentage cured was 88.3%.

**Temporal profiles of urine and serum IgGs post-treatment in cured group.** The participants for the follow-up study (F/U) were rearranged into three groups. The F/U 1 group (cured group) consisted of individuals who were *S. stercoralis*-positive by fecal examination or ddPCR at baseline screening and were free from infection at three months post-treatment (n = 121). Among them, 113 were positive by urine IgG and 105 by serum IgG (F/U 1). The second group (F/U 2) consisted of 29 out 169 immunologically low responder participants who were positive for *S. stercoralis* by fecal examination or ddPCR but were seronegative by urine IgG (15 out of 29) and serum IgG-ELISA (20 out of 29) at baseline pre-treatment. The third group (F/U 3) was the parasite-negative group comprising individuals who were parasitological and serological negatives at baseline (n = 40). In all three F/U groups, the levels of IgGs in urine and serum were monitored at baseline and at 3, 6, 9, 12, and 18 months post-treatment. After 3 months post-treatment, fecal examination and/or ddPCR positive were excluded from analysis in all three F/U groups.

In F/U 1, the concentrations of *Strongyloides*-specific IgG antibody in urine and serum between baseline (pre-treatment) and post-treatment significantly decreased at months 3–18 (Wilcoxon matched-pairs test, *p* value < 0.0001) (Table 5 and S3 Fig). The positive detection rates of *S. stercoralis* by urine and serum IgG-ELISAs were also significantly decreased at months 3–18 post-treatment (Chi-square test for trend = 27.5–151.7, *p* value < 0.0001) (Fig 3A and 3B). The lowest proportion of individuals who were diagnosed as positive by IgG was at 12 months post-treatment (34.6% for urine IgG and 10% for serum IgG). The temporal profiles of concentrations of *Strongyloides*-specific IgG in urine and serum also showed a significant reduction (Kruskal-Wallis test = 113.4–212.6, *p* value < 0.0001) (Fig 3C and 3D). The trends of serum IgG showed more consistent decline with time post treatment than that of urine IgG.

**Table 5. Comparisons of *Strongyloides*-specific IgG in serum and urine of cured group between pre-treatment baseline and 3–18 months post-IVM treatment.**

| Intervals post-treatment (months) | *Strongyloides*-specific IgG unit [mean ± SE, $\log_{10}(X+1)$] | | | | | |
|---|---|---|---|---|---|---|
| | Urine IgG | | | Serum IgG | | |
| | n[a] | Pre-treatment | Post-treatment | n[a] | Pre-treatment | Post-treatment |
| 3 | 113 | 2.96 ± 0.05 | 2.32 ± 0.05 | 105 | 3.00 ± 0.04 | 2.63 ± 0.04 |
| 6 | 95 | 2.96 ± 0.05 | 2.31 ± 0.05 | 86 | 3.01 ± 0.05 | 2.41 ± 0.04 |
| 9 | 78 | 2.92 ± 0.06 | 2.36 ± 0.05 | 69 | 3.03 ± 0.05 | 2.06 ± 0.05 |
| 12 | 26 | 3.30 ± 0.08 | 2.10 ± 0.11 | 20 | 2.88 ± 0.10 | 1.82 ± 0.07 |
| 18 | 42 | 2.72 ± 0.08 | 2.24 ± 0.08 | 40 | 3.04 ± 0.06 | 2.18 ± 0.05 |

[a] total number of the same participants at baseline and post treatment.

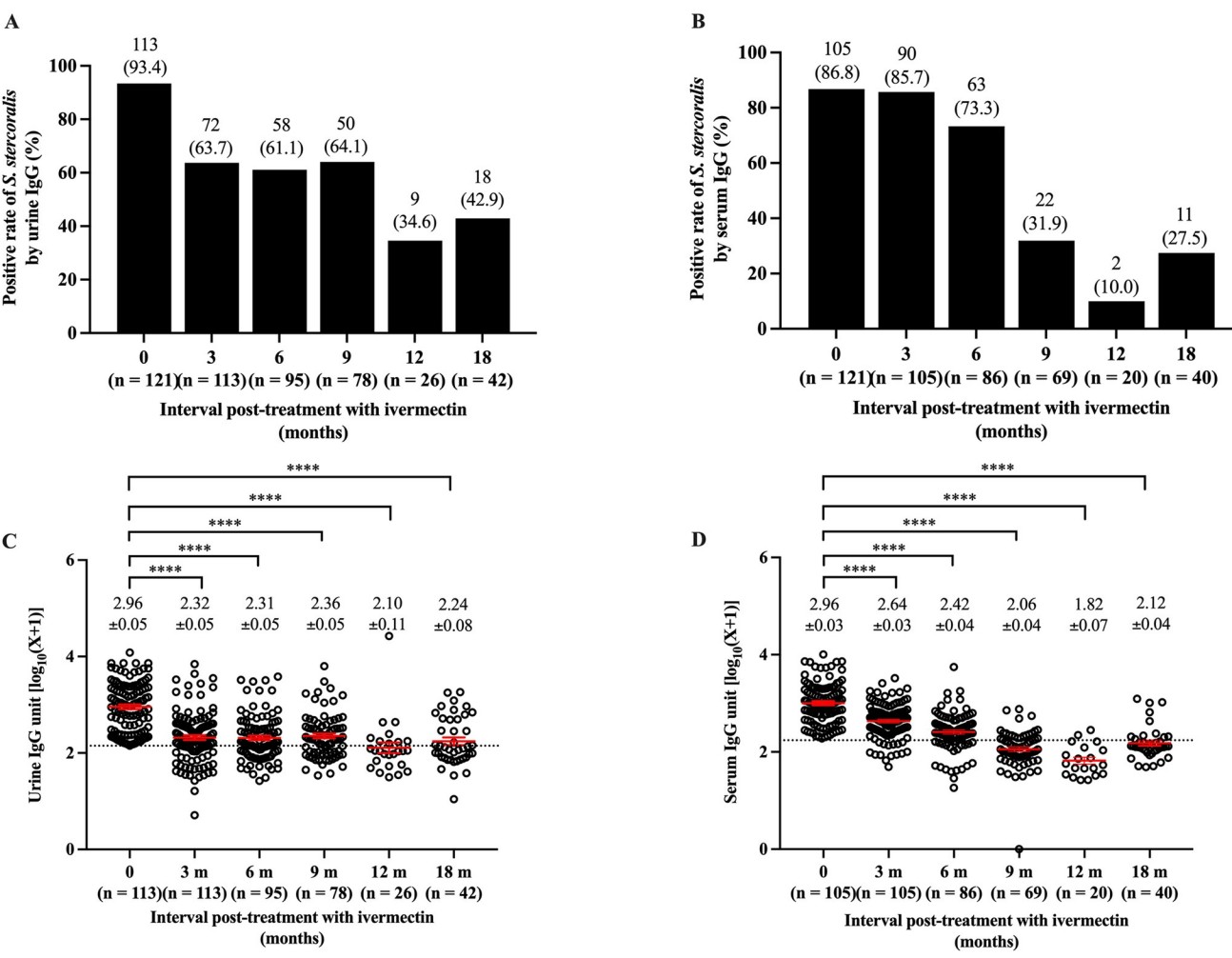

**Fig 3. The temporal profiles of positive rates and levels of IgG of cured group at pre-treatment baseline and post-treatment (3–18 months).** Positive detection rates of *Strongyloides*-specific IgG in urine (A) and serum (B). Concentrations of IgG in urine (C) and serum (D). Dots are individual values of IgG, and the horizontal lines are mean ± SE of IgG antibody units/ml of urine ($\log_{10}[X+1]$). The IgG cutoff values ($\log_{10}[X+1]$) were 2.15 for urine and 2.24 for serum. ****Statistical tests indicated $p$ value < 0.001.

Analysis based on urine IgG measurement exhibited a slight upward trend of treatment response from 77.0% to 100% from months 3 to 12 and reduced to 66.7% at month 18. However, there was no statistically significant association between positive rates and time post-treatment (Chi-square test for trend = 0.30, $p$ value > 0.05). In the case of serum IgG, treatment response started at 59% at month 3, increased to 76.6% at month 6, and ascended to 97–100% at months 9 and 12 (Chi-square test for trend = 26.11, $p$ value < 0.0001) (Fig 4A). The conversion to negative rates by urine IgG steadily increased from 36.3% at three months post-treatment and were the highest (65.4%) at month 12. The conversion to negative rates were significantly positively associated with time post-treatment (Chi-square test for trend = 7.57, $p$ value < 0.01). Based on serum IgG, the rates were also positively associated with time post-treatment (Chi-square test for trend = 78.63, $p$ value < 0.0001) (Fig 4B).

**Temporal profiles of urine and serum IgGs in low responder and parasite-negative groups.** Among the low responders (F/U 2), the *Strongyloides*-specific IgG levels (unit/mL)

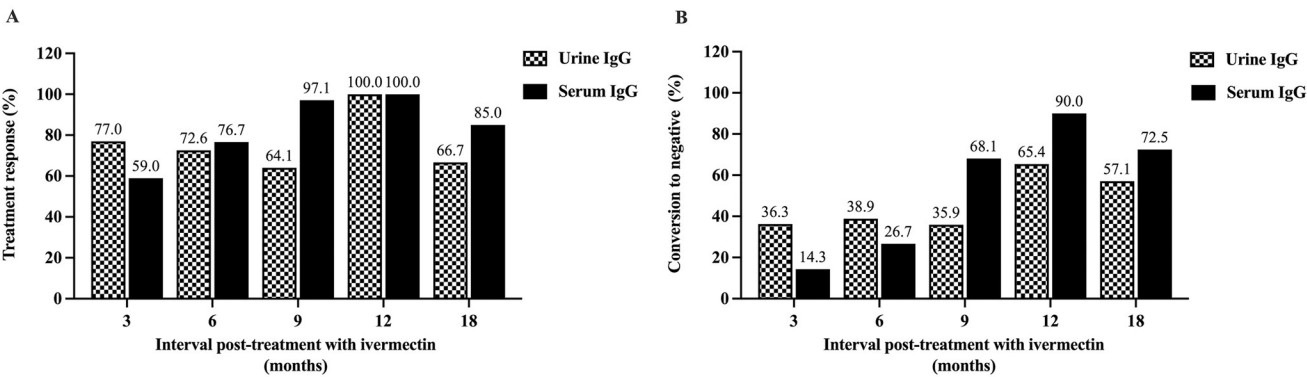

**Fig 4. Treatment response and conversion to negative post-IVM treatment of cured group.** Treatment response rates determined by *Strongyloides*-specific IgG in urine and serum (A). The conversion to negative rates determined by *Strongyloides*-specific IgG in urine and serum (B). The IgG cutoff values ($\log_{10}[X+1]$) were 2.15 for urine and 2.24 for serum.

in serum and urine were generally lower than the cutoff values during the 18 months study period (Kruskal-Wallis test = 9.68–10.20, $p$ value > 0.05). However, seven IgG positive cases were detected, i.e., serum IgG at months 3, 6 and 18 and urine IgG at 9 and 18 months (S4A and S4B Fig). Among these positive IgG cases, their corresponding fecal examination and ddPCR were negative for *S. stercoralis*.

Within the parasite-negative group (F/U 3 group), the levels of *Strongyloides*-specific urine IgG were relatively stable at the various sampling times (Kruskal-Wallis test = 7.787, $p$ value > 0.05) (S5A Fig). Meanwhile, the levels of serum IgG had slight variations (Kruskal-Wallis test = 12.52, $p$ value < 0.01) but were lower than the cutoff value (S5B Fig).

## Discussion

At baseline study, urine and serum IgG ELISAs in diagnosing strongyloidiasis showed high sensitivity and high concordance between methods which were consistent with the previous reports [14, 32]. The observed age-prevalence pattern of strongyloidiasis in this study also indicated that age as risk factors for *S. stercoralis* infection in indigenous populations. Thus in endemic communities with strongyloidiasis, the prevalence and infection intensity in adults were significantly higher than in young or children [13, 32–35].

In this study, the prevalence of *S. stercoralis* infection by the combined fecal examination and ddPCR was 48.1%, significantly lower than that of urine and serum IgG-ELISAs (70.7% and 68.4%, respectively). The high *S. stercoralis* prevalences by fecal examination were also reported in several neighbouring endemic areas. Up to 61% prevalence was reported in different endemic areas in Thailand [3], 33.4–44.2% in Lao PDR [36–38], and 44.7% in Cambodia [34]. Similarly, a high strongyloidiasis seroprevalence of 58.9–65.1% by urine IgG-ELISA was reported in northeast Thailand [13] and in Cambodia (88.9%) [39]. As per the findings of the present study, co-infections of individuals with *Strongyloides* and other parasites, especially *O. viverrini*, were frequently reported in northeast Thailand [18, 21, 40, 41]. The high prevalence of strongyloidiasis in Thailand and the neighboring countries, signaling high transmission rates, and a high prevalence of co-infection with other helminths indicate a need for an improved strategy for accurate diagnosis, treatment and monitoring, and control of strongyloidiasis.

Parasitological methods for strongyloidiasis have a low sensitivity for diagnosis and evaluating drug treatment, especially to exclude cryptic asymptomatic infections. Therefore, high-

performance diagnostic methods such as PCR and serology are clear alternatives. Thus, in addition to fecal examination, we applied the ddPCR to detect *S. stercoralis* DNA in fecal samples and assess the efficacy of IVM treatment at 3 months post treatment. The result showed that the cure rate of *S. stercoralis* infection by IVM treatment was 88.3%. Three months post-treatment, the ddPCR was still positive in 10 out of 131 individuals (7.6%) in larval-negative participants. Thus, ddPCR was able to detect some cases of IVM treatment failure in human strongyloidiasis. Comparable cure rates of 82–84% and 85–86% based on worm clearance and antibody reduction by single and multiple doses of IVM administration have been previously reported [22]. The question then arises whether the treatment regimen given in this study (one dose of 200 μg/kg IVM) should be revised to increase the cure rate. Alternatively, whether the report of positive detection of *Strongyloides*-specific DNA in negative fecal examination after IVM treatment reflects the poor efficacy of IVM or not is unclear [42].

Another approach for evaluating the response to drug treatment is looking at the time post-treatment when there is a significant decrease of antibodies and when seroconversion happens. The antibody should decrease at least half of the initial levels, while seroconversion or the conversion to negative is the transition from positive to negative results [16]. In the present study, IgG concentration in urine and serum significantly decreased from month 3 and became the lowest at month 12 post-treatment, similar to a previous report [16]. We noted a ~30% reduction in positive detection rate in urine IgG compared to baseline values at months 3 to 9, and it was more rapid than the reduction in positive detection rate in serum IgG. The latter showed a non-significant reduction in month three and a small percentage reduction in months 6 and 9.

The persistence of positive IgG in some individuals throughout the study period in this study with the positive rates of 27.5% for urine IgG and 42.9% for serum IgG at the end of the 18 month study period is puzzling. Similar results were reported that serum IgG and IgG4 antibody levels significantly declined one year after treatment, but these antibodies persisted at all tested time points, including at more than one year post-treatment [16, 43–48]. Whether these positives IgG detections are consequence of partial worm clearance from drug treatment and/or reactivation of tissue larvae (L3a) deserve more study [49–51].

Our study suggested that as short as three months post-treatment is a suitable time to measure IgG when monitoring treatment response were 77.0% by urine IgG and 36.6% by serum IgG. The treatment response measured (antibody ratio of baseline and post treatment > 0.6) was 38.4% at 3 months and further increased to 91.7% at 18 months after treatment [43]. The treatment responses at 12 months post treatment varied from 81.2–89.8% by ELISAs [16]. In our study, the time profiles of the conversion to negative rates by urine and serum IgGs were similar, but the rates by serum IgG were slightly higher than urine IgG in months 9 to 18. The conversion to negative rates peaked at 65.4% by urine IgG and 90.0% by serum IgG at 12 months post treatment. Compared with previous studies, conversion to negative (negativization) by serum IgG varied between 54.5–72.5% by different types of ELISA [16].

To monitor the efficacy of drug treatment in the low responder group, fecal examination by APCT, FECT and PCR proved useful in this study. Within these low responders who had low IgG antibody concentrations in urine (6 out of 15 participants), it is unknown whether removal of water by concentrating urine may contribute to higher detection rates [52].

We acknowledged that there were several limitations in this long-term diagnostic study. First, the ddPCR was performed using samples at selected time points to confirm *S. stercoralis* infection. If more or other time points were collected, it might affect the treatment response and conversion to negative rates. Second, treatment response based on the urine and serum IgG ELISA was not applicable in low responder group (F/U group 2), hence the parasitological or DNA detection of *S. stercoralis* is needed. A third limitation is that only one stool sample was collected in assessing the cure rate, thus reducing the sensitivity of fecal detection [32, 49].

Thus, some seemingly cured individuals may be positive if sampled multiple times on different days. However, including the highly sensitive molecular detection method has partially addressed this limitation. Lastly, similar study approach should be conducted in other endemic populations with low prevalence of *S. stercoralis* and varying coexisted soil-transmitted helminthiasis and taeniasis.

## Conclusion

Our findings suggest that detecting urine IgG by ELISA showed similar performance in sensitivity to serum IgG-ELISA. Thus, the former can be an alternative diagnostic tool (to serum IgG-ELISA) for human strongyloidiasis. The follow-up study also showed that urine and serum IgG can be used to evaluate the efficacy of IVM treatment of *S. stercoralis* infection in terms of treatment responses and negavitization. Since urine sample collection is non-invasive and much more convenient to handle than blood samples, urine IgG-ELISA can be used to assess treatment response and public health control programs against strongyloidiasis.

## Supporting information

**S1 Fig. Prevalence and intensity of *S. stercoralis* infection classified by gender using fecal examination and antibody detection in urine and serum.** The prevalence of *S. stercoralis* infection classified by gender by all methods (A). Intensity by fecal examination (B), urine IgG-ELISA (C) and serum IgG-ELISA (D). The graphs with box whisker plots represented min-max with median and + symbol represented mean value.
(TIF)

**S2 Fig. Correlation between levels of *Strongyloides*-specific IgG in urine and serum.** Data shown are observed values and the solid line was calculated from the regression equation (Y = ax+b, Y = log IgG, a = slope, x = log IgG, b = Y-intercept). Dotted lines (vertical and horizontal) represent the cutoff values.
(TIF)

**S3 Fig. Individual *Strongyloides*-specific IgG levels of 'cured group' at pre-treatment baseline and post-treatment (3–18 months).** *Strongyloides*-specific IgG in urine (A-E) and in serum (F-J). Significant different between pair samples by Wilcoxon matched-pairs test, **** *p* value < 0.0001.
(TIF)

**S4 Fig. Temporal profiles of *Strongyloides*-specific IgG post-IVM treatment in in urine and serum of low responder group.** *Strongyloides*-specific IgG antibody units in urine (A) and serum (B). Data points were individual values and error bar represents mean ± SE of IgG antibody units/ml ($\log_{10}[X+1]$).
(TIF)

**S5 Fig. Temporal profiles of *Strongyloides*-specific IgG in urine and serum of endemic parasite-negative group.** *Strongyloides*-specific IgG in urine (A) and serum (B), respectively. Error bar represents mean ± SE of IgG antibody units/ml ($\log_{10}[X+1]$).
(TIF)

**S1 File. Raw data for data analysis and graph construction at pre-treatment (baseline) and post-treatment.**
(XLSX)

**S1 Checklist. Human participants research checklist.**
(DOCX)

**S2 Checklist.**
(DOCX)

# Acknowledgments

We would like to thank the participants in Phon Ngam subdistrict, Kamalasai district, Kalasin province, for their support and collaboration throughout the project duration.

# Author Contributions

**Conceptualization:** Phattharaphon Wongphutorn, Chatanun Eamudomkarn, Paiboon Sithithaworn.

**Data curation:** Phattharaphon Wongphutorn.

**Formal analysis:** Phattharaphon Wongphutorn.

**Funding acquisition:** Paiboon Sithithaworn.

**Investigation:** Kulthida Y. Kopolrat, Chatanun Eamudomkarn, Nuttanan Hongsrichan, Opal Pitaksakulrat, Jiraporn Sithithaworn, Patcharaporn Tippayawat, Anchalee Techasen, Rahmah Noordin, Paiboon Sithithaworn.

**Methodology:** Phattharaphon Wongphutorn, Chatanun Eamudomkarn, Paiboon Sithithaworn.

**Project administration:** Phattharaphon Wongphutorn, Paiboon Sithithaworn.

**Resources:** Kulthida Y. Kopolrat, Chanika Worasith.

**Supervision:** Paiboon Sithithaworn.

**Validation:** Phattharaphon Wongphutorn, Paiboon Sithithaworn.

**Writing – original draft:** Phattharaphon Wongphutorn.

**Writing – review & editing:** Chatanun Eamudomkarn, Rahmah Noordin, Thomas Crellen, Paiboon Sithithaworn.

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
