## [Decision Letter · Decision Letter 0]

16 Sep 2024

PONE-D-24-24002Diagnostic value of urinary and serum IgG antibodies in evaluating drug treatment response in strongyloidiasis assessed by fecal examination and digital droplet PCRPLOS ONE

Dear Dr. Sithithaworn,

Thank you for submitting your manuscript to PLOS ONE. After careful consideration, we feel that it has merit but does not fully meet PLOS ONE’s publication criteria as it currently stands. Therefore, we invite you to submit a revised version of the manuscript that addresses the points raised during the review process. Please respond to each of the two external reviewers' comments and revise the manuscript accordingly.

We look forward to receiving your revised manuscript.

Kind regards,

David Joseph Diemert, M.D.

Academic Editor

PLOS ONE

Journal Requirements:

“1.National Research Council of Thailand (NRCT)

2.Basic Research Fund of Khon Kaen University through Cholangiocarcinoma Research Institute (CARIBRF64-50)”

4. We note that your Data Availability Statement is currently as follows: [All relevant data are within the manuscript and its Supporting Information files]

Reviewers' comments:

Reviewer's Responses to Questions

**Comments to the Author**

1. Is the manuscript technically sound, and do the data support the conclusions?

Reviewer #1: Partly

Reviewer #2: Yes

2. Has the statistical analysis been performed appropriately and rigorously? 

Reviewer #1: Yes

Reviewer #2: I Don't Know

3. Have the authors made all data underlying the findings in their manuscript fully available?

Reviewer #1: No

Reviewer #2: Yes

4. Is the manuscript presented in an intelligible fashion and written in standard English?

Reviewer #1: Yes

Reviewer #2: Yes

5. Review Comments to the Author

Reviewer #1: Diagnostic value of urinary and serum IgG antibodies in evaluating drug treatment response in strongyloidiasis assessed by fecal examination and digital droplet PCR

A lot of data make it a bit difficult to follow.

I would like to see the results in the same order as in the M&M section with the same paragraph haedings.

Line 258 - 265 please provide the actual numbers for the ddPCR, APCT and composite standard as well likewise as in table 1 for FECT

Line 233 The IgG cutoff values (antibody units/ml) were 140.7 in urine and 174.4 in serum. Seems in contrast with the cut-off line in figures 3C and 3D just above 2

Comparing fig3C and fig3D, it seems to me that serum IgG follows a more logical decline in time as compared to urine IgG

Reviewer #2: This manuscript reports the results of study conducted in a cohort of participants with and without strongyloidiasis to compare the performance of an in-house ELISA developed (and previously published) that utilises Strongyloides ratti larval antigens to detect IgG in urine or serum. The authors have used, as a composite diagnostic reference test, a combination of microscopy + agar culture and/or ddPCR detection of parasite DNA in stools. The study population, residents of a region endemic for strongyloidiasis, with a previously reported prevalence of >20%, included sub-groups of ‘proven’ Strongyloides-infected, Strongyloides-uninfected but infected with other helminth/non-helminth parasites, and uninfected individuals. In addition to a direct comparison of the three diagnostic tests in baseline (pre-treatment) samples, a sub study compared decline in the Strongyloides-specific IgG levels in urine and serum at 5 pre-selected timepoints post treatment with a single dose of ivermectin.

The experimental design and study populations are credible, and the study appears to have been conducted in a planned and methodical manner. The key findings of the study are, first, a high prevalence of strongyloidiasis (48%) in the screened, selected population, which may be explained by the selection criteria used in enrolment of the study populations. Second, the sensitivity of the ELISA when applied to urine was high (91%), essentially equivalent or higher than that of the serum assay (88%), as was the accuracy (measured by AUC of ROC: ~0.66 for urine and ~0.73 for serum) when compared with the reference composite tests. Third, there was a moderate-strong correlation between urine and serum IgG positivity. Fourth, the results of the post-treatment follow up is in line with previous studies showing a decline in IgG to Strongyloides Ags by 3 months post treatment.

These findings are remarkable and portend a useful role for the urine assay in the diagnostic “toolbox” for strongyloidiasis, but these preliminary findings in a selected population will need confirmation in a less selected populations, including in populations with lower prevalence rates of Strongyloides infection and those with other helminth infections known to cross react with Strongyloides Ags (primarily other soil-transmitted helminths and tapeworms).

Specific comments

Methods

- Ethics: were the participants in the treatment-follow up cohorts blinded to the category to which they were assigned? Could this knowledge affect their exposure to re-infection (e.g. by changing behaviour that risks exposure)?

- The term ‘proven’ positive/negative is somewhat misleading – it presumably refers to the reference test positive individuals – samples that were positive on either /and culture + microscopy or PCR, i.e., parasite-detection tests. I would recommend using ‘reference positive’ or ‘Strongyloides positive by reference test’

- ‘Negativization’ is an awkward and grammatically unconventional term – consider a better term, maybe something like ‘Conversion to negative’ or ‘reversion to negative’.

- Statistical analysis: although most readers will assume that corrections for multiple comparisons are incorporated into the p-values shown, it should be stated specifically whether this type of correction was performed in the analyses

Results

- Table 1 and lines 250-256: the authors have listed the numbers of those infected with S. stercorales (Ss) and with other parasites. Do those that are indicated as having Ss infection have only Ss in their stool? How many had both Ss and other parasites?

- Since the denominator used to calculate the percentages for detection rates (Table 2) and sensitivity (Table 3) is the number positive by the reference test (i.e., ‘True positives’), to avoid confusing readers who are used to the conventional 2 x2 cross-tabulation format of gold-standard test and experimental test, this should be so stated in the text or legend to the two tables.

- Table 3: Please check the means for sensitivity for all the cells – e.g, the mean for urine (91.1) falls outside the CI95 (98.8-95)

- Figures 3 and 4: The legend should include a description of what was the definition of ‘positive’ and ‘negative’ for urine and serum. In Figure 3/4, the X-axis title could be better expressed as ‘Interval post-treatment with ivermectin’

6. PLOS authors have the option to publish the peer review history of their article (what does this mean?). If published, this will include your full peer review and any attached files.

Reviewer #1: No

Reviewer #2: No

---

## [Author Response · Author response to Decision Letter 0]

26 Sep 2024

Response to Reviewer

1. Reviewer #1 (Green highlight)

1.1 A lot of data make it a bit difficult to follow. I would like to see the results in the same order as in the M&M section with the same paragraph headings.

Answer: Thank you for the comment. We realized this point while preparing the manuscript and that is why some results are separated as supporting information such that the readers can easily follow the main results of the manuscript. 

In respond to the comment, the headings in the result section were modified by adding “Baseline study” (line 252) to cover the baseline study results and is clearly separated from the Follow-up study. The subheading for prevalence and intensity were combined to the flow of the results (line 268-302). 

1.2 Line 258 - 265 please provide the actual numbers for the ddPCR, APCT and composite standard as well likewise as in table 1 for FECT

Answer: We revised the text as “The prevalence of S. sterocalis determined by APCT was 45.8% (124/271), 21.4% (75/351) by FECT and, 45.3% (43/95) by ddPCR. The prevalence by composite reference tests (combined APCT, FECT and ddPCR) was 48.1% (169/351) (line no.255-258). More relevant data are also given Table 1.

1.3 Line 233 The IgG cutoff values (antibody units/ml) were 140.7 in urine and 174.4 in serum. Seems in contrast with the cut-off line in figures 3C and 3D just above 2

Answer: The IgG cutoff values (antibody units/ml) were log-transformed values [log10 (unit+1)] of 2.15 and 2.24 for IgG in urine and serum, respectively (line no. 233-235, Fig 3 and 4 legends).

1.4 Comparing fig3C and fig3D, it seems to me that serum IgG follows a more logical

decline in time as compared to urine IgG

Answer: We added a sentence “The trends of serum IgG showed more consistent decline with time post treatment than that of urine IgG” (line no.370-371). 

2. Reviewer #2 (Gray highlight)

This manuscript reports the results of study conducted in a cohort of participants with and without strongyloidiasis to compare the performance of an in-house ELISA developed (and previously published) that utilises Strongyloides ratti larval antigens to detect IgG in urine or serum. The authors have used, as a composite diagnostic reference test, a combination of microscopy + agar culture and/or ddPCR detection of parasite DNA in stools. The study population, residents of a region endemic for strongyloidiasis, with a previously reported prevalence of >20%, included sub-groups of ‘proven’ Strongyloides-infected, Strongyloides-uninfected but infected with other helminth/non-helminth parasites, and uninfected individuals. In addition to a direct comparison of the three diagnostic tests in baseline (pre-treatment) samples, a sub study compared decline in the Strongyloides-specific IgG levels in urine and serum at 5 pre-selected timepoints post treatment with a single dose of ivermectin. The experimental design and study populations are credible, and the study appears to have been conducted in a planned and methodical manner. The key findings of the study are, first, a high prevalence of strongyloidiasis (48%) in the screened, selected population, which may be explained by the selection criteria used in enrolment of the study populations. Second, the sensitivity of the ELISA when applied to urine was high (91%), essentially equivalent or higher than that of the serum assay (88%), as was the accuracy (measured by AUC of ROC: ~0.66 for urine and ~0.73 for serum) when compared with the reference composite tests. Third, there was a moderate-strong correlation between urine and serum IgG positivity. Fourth, the results of the post-treatment follow up is in line with previous studies showing a decline in IgG to Strongyloides Ags by 3 months post treatment.

These findings are remarkable and portend a useful role for the urine assay in the diagnostic “toolbox” for strongyloidiasis, but these preliminary findings in a selected population will need confirmation in a less selected populations, including in populations with lower prevalence rates of Strongyloides infection and those with other helminth infections known to cross react with Strongyloides Ags (primarily other soil-transmitted helminths and tapeworms).

Answer: We agreed with the suggestion and this matter was added to the Discussion section text “Lastly, similar study approach should be conducted in other endemic populations with low prevalence of S. stercoralis and coexisting soil-transmitted helminthiasis and taeniasis (line 494-496). 

3. Specific comments (yellow highlight)

3.1 Methods

3.1.1 Ethics: were the participants in the treatment-follow up cohorts blinded to the category to which they were assigned? Could this knowledge affect their exposure to re-infection (e.g. by changing behavior that risks exposure)?

Answer: The population category of treatment-follow up study was blinded among the project participants and the diagnostic results were kept private issue. We did not advice the participants for the risk of exposure to S. stercoralis infection during the study duration. At the end of the project, a comprehensive prevention and control education as well as drug treatment (as needed) were given to all participants according to the human ethical procedure. 

3.1.2 The term ‘proven’ positive/negative is somewhat misleading – it presumably refers to the reference test positive individuals – samples that were positive on either /and culture + microscopy or PCR, i.e., parasite-detection tests. I would recommend using ‘reference positive’ or ‘Strongyloides positive by reference test’

Answer: We have changed “proven strongyloidiasis” to “Strongyloides positive by reference test” as suggested throughout the manuscript text and in Table 2.

3.1.3 ‘Negativization’ is an awkward and grammatically unconventional term – 

consider a better term, maybe something like ‘Conversion to negative’ or ‘reversion to 

negative’.

Answer: We changed the word “negativization” to “conversion to negative” throughout the manuscript.

3.1.4 Statistical analysis: although most readers will assume that corrections for

multiple comparisons are incorporated into the p-values shown, it should be stated specifically whether this type of correction was performed in the analyses.

Answer: Thank you for raising this point. We added the sentences in Statistical analysis as “The time profiles of positive rates and levels of Strongyloides-specific IgG in urine and serum were validated by Chi square test for trend and Kruskal-Wallis test, respectively. The corrections to the Type I error for multiple comparison was implemented to ensure the confidence in the reported results. (line no.246-250).

3.2 Results

3.2.1 Table 1 and lines 250-256: the authors have listed the numbers of those infected with S. stercorales (Ss) and with other parasites. Do those that are indicated as having Ss infection have only Ss in their stool? How many had both Ss and other parasites?

Answer: We added the requested data in the revised Table 1.

3.2.2 Since the denominator used to calculate the percentages for detection rates (Table 2) and sensitivity (Table 3) is the number positive by the reference test (i.e., ‘True positives’), to avoid confusing readers who are used to the conventional 2 x2 cross-tabulation format of gold-standard test and experimental test, this should be so stated in the text or legend to the two tables.

Answer: We added details of a composite reference standard in table 2 and 3 legend as suggested. 

3.2.3 Table 3: Please check the means for sensitivity for all the cells – e.g, the mean for urine (91.1) falls outside the CI95 (98.8-95)

Answer: We revised the 95% CI for sensitivity to be 85.8%-95.0% in Table 3.

3.2.4 Figures 3 and 4: The legend should include a description of what was the definition of ‘positive’ and ‘negative’ for urine and serum. In Figure 3/4, the X-axis title could be better expressed as ‘Interval post-treatment with ivermectin’

Answer: We have clarified the cutoff values as “The definition of ‘positive’ and ‘negative’ for urine and serum by the IgG cutoff values (log10[X+1]) which was 2.15 for urine and 2.24 for serum (Fig 3 and 4 legends). The X-axis title in Fig 3 and 4 were revised to ‘Interval post-treatment with ivermectin’ in Fig 3 and 4.

---

## [Decision Letter · Decision Letter 1]

14 Nov 2024

Diagnostic value of urinary and serum IgG antibodies in evaluating drug treatment response in strongyloidiasis assessed by fecal examination and digital droplet PCR

PONE-D-24-24002R1

Dear Dr. Sithithaworn,

We’re pleased to inform you that your manuscript has been judged scientifically suitable for publication and will be formally accepted for publication once it meets all outstanding technical requirements.

Kind regards,

David Joseph Diemert, M.D.

Academic Editor

PLOS ONE

Additional Editor Comments (optional):

Reviewers' comments:

Reviewer's Responses to Questions

**Comments to the Author**

1. If the authors have adequately addressed your comments raised in a previous round of review and you feel that this manuscript is now acceptable for publication, you may indicate that here to bypass the “Comments to the Author” section, enter your conflict of interest statement in the “Confidential to Editor” section, and submit your "Accept" recommendation.

Reviewer #1: All comments have been addressed

Reviewer #2: All comments have been addressed

2. Is the manuscript technically sound, and do the data support the conclusions?

Reviewer #1: Yes

Reviewer #2: Yes

3. Has the statistical analysis been performed appropriately and rigorously? 

Reviewer #1: Yes

Reviewer #2: Yes

4. Have the authors made all data underlying the findings in their manuscript fully available?

Reviewer #1: No

Reviewer #2: Yes

5. Is the manuscript presented in an intelligible fashion and written in standard English?

Reviewer #1: Yes

Reviewer #2: Yes

6. Review Comments to the Author

Reviewer #1: PONE-D-24-24002 Diagnostic value of urinary and serum IgG antibodies in evaluating drug treatment response in strongyloidiasis assessed by fecal examination and digital droplet PCR

Well written and very elaborate work, three things that would have added value

- A figure showing the positive findings in either FECT, APCT and/or ddPCR

- Quantification with FECT is given, please also give quantification with ddPCR

- Figures for urine and serum IgG like S3fig but all time points in one figure can replace S3fig

Reviewer #2: The revisions to the manuscript addressed my questions adequately and I believe that the changes have improved it.

7. PLOS authors have the option to publish the peer review history of their article (what does this mean?). If published, this will include your full peer review and any attached files.

Reviewer #1: No

Reviewer #2: No

---

## [Editor Report · Acceptance letter]

20 Nov 2024

PONE-D-24-24002R1 

PLOS ONE

Dear Dr. Sithithaworn, 

I'm pleased to inform you that your manuscript has been deemed suitable for publication in PLOS ONE. Congratulations! Your manuscript is now being handed over to our production team.

Kind regards, 

on behalf of

Dr. David Joseph Diemert 

Academic Editor

PLOS ONE